# Diabetes Mellitus and Dental Implants: A Systematic Review and Meta-Analysis

**DOI:** 10.3390/ma15093227

**Published:** 2022-04-29

**Authors:** Yasmin Al Ansari, Halime Shahwan, Bruno Ramos Chrcanovic

**Affiliations:** 1Faculty of Odontology, Malmö University, 214 21 Malmo, Sweden; yasminalansari98@gmail.com (Y.A.A.); halimeshahwan@hotmail.com (H.S.); 2Department of Prosthodontics, Faculty of Odontology, Malmö University, 214 21 Malmo, Sweden

**Keywords:** dental implant, failure, marginal bone loss, diabetes mellitus, systematic review, meta-analysis, meta-regression

## Abstract

The present review aimed to evaluate the impact of diabetes mellitus on dental implant failure rates and marginal bone loss (MBL). An electronic search was undertaken in three databases, plus a manual search of journals. Meta-analyses were performed as well as meta-regressions in order to verify how the odds ratio (OR) and MBL were associated with follow-up time. The review included 89 publications. Altogether, there were 5510 and 62,780 implants placed in diabetic and non-diabetic patients, respectively. Pairwise meta-analysis showed that implants in diabetic patients had a higher failure risk in comparison to non-diabetic patients (OR 1.777, *p* < 0.001). Implant failures were more likely to occur in type 1 diabetes patients than in type 2 (OR 4.477, *p* = 0.032). The difference in implant failure between the groups was statistically significant in the maxilla but not in the mandible. The MBL mean difference (MD) between the groups was 0.776 mm (*p* = 0.027), with an estimated increase of 0.032 mm in the MBL MD between groups for every additional month of follow-up (*p* < 0.001). There was an estimated decrease of 0.007 in OR for every additional month of follow-up (*p* = 0.048). In conclusion, implants in diabetic patients showed a 77.7% higher risk of failure than in non-diabetic patients.

## 1. Introduction

Diabetes mellitus is a group of metabolic diseases characterized by hyperglycemia (high levels of glucose in the blood) which results from defects in insulin secretion (the pancreas does not produce enough insulin), insulin action (the body cannot effectively use the insulin it produces), or both [1]. The most common type of diabetes mellitus, type 2, which accounts for 90–95% of those with diabetes mellitus [1], was estimated to affect 537 million adults worldwide in 2021, with a prediction to rise to 643 million adults by 2030 [2]. Such prevalence highlights the importance of this group of diseases.

The long-term hyperglycemia of diabetes mellitus very commonly leads to failure, damage, and/or dysfunction of many tissues and organs of the human body, causing substantial clinical morbidity [1,3]. Moreover, the duration of diabetes may impact the clinical and functional status of the individuals, a factor that is suggested to be independent of glycemic control and age [4]. These consequences usually result from a set of negative effects of the disease, which include delayed wound healing [5], microvascular complications [6], impaired response to infection [7], impaired bone metabolism, and bone strength [8], among others. For individuals who have onset of type 2 diabetes in youth, the risk of microvascular and other complications increases steadily over time and affects most individuals by the time of young adulthood [9].

Glycemia, the level of sugar in the blood, may play an important role in these consequences, as a correlation between glycemic control and the development of microvascular and macro-vascular complications was observed [10]. Tight and intensive glycemic control in diabetic patients can delay the onset and the progression of many microvascular-related complications associated with the condition [11], although the effects of this control seem to become weaker once complications have been manifested [12]. A controlled diabetic patient is defined as a patient that keeps their glycemia as close to normal as possible. This is established by a test, which measures what percentage of hemoglobin proteins in the blood are coated with sugar, namely what percentage of hemoglobin is glycated (HbA1c). Diabetic individuals that keep a level up to 6.5% HbA1c are considered patients with controlled diabetes mellitus [13].

The negative effect of the disease on bone metabolism has raised some concerns about the long-term survival of dental implants in diabetic patients. A previous systematic review on the subject had shed some light on the issue [14]. The results suggested that diabetes mellitus does exert an influence on the implant failure rates when compared to non-diabetic patients. However, this previous review is based on only 14 studies. It was, therefore, the aim of the present systematic review to compare the implant failure rates and marginal bone loss (MBL) between diabetic and non-diabetic patients in an update of the previous study.

## 2. Materials and Methods

This study followed the PRISMA 2020 Statement guidelines [15]. The review was registered in PROSPERO (CRD42021240670).

### 2.1. Objective

The purpose of the present study was to test the null hypothesis of no difference in the implant failure rates and MBL after the insertion of dental implants in diabetic patients compared to the insertion in non-diabetic patients against the alternative hypothesis of a difference based on a systematic review of the literature.

The focused question was elaborated by using the PICO format (participants, interventions, comparisons, outcomes): In partially and fully edentulous patients (participants) being rehabilitated with dental implants (intervention), is there a difference between diabetic and non-diabetic patients (comparison) on implant failure rates and MBL (outcomes)?

### 2.2. Search Strategies

An electronic search without time restrictions was undertaken and last updated in October 2021 in the following databases: PubMed/Medline, Web of Science, and Scopus. The following terms were used in the search strategies:

(“dental implant” OR “oral implant”) AND (diabetes OR diabetic)

A manual search of dental implant-related journals (listed in the Appendix A) was performed. The reference list of the identified studies and the relevant reviews on the subject were also checked for possible additional studies.

### 2.3. Inclusion and Exclusion Criteria

Clinical human studies were included, with information on implant failure rates in diabetic and non-diabetic individuals rehabilitated with cylindrical modern dental implants of commercially pure titanium or its alloys. As an individual is either diabetic or not, it is impossible to randomize the placement of implants for this condition. Therefore, non-randomized and retrospective clinical studies were also considered for inclusion in the present review.

Only studies including diabetic patients under glycemic control were included. This information, when not available in the publications, was obtained by contact with the authors of the articles.

Case reports, technical reports, animal and in vitro studies, and review papers were excluded. Studies evaluating mini-implants, zygomatic, orthodontic, zirconia, subperiosteal, or hollow implants were excluded.

### 2.4. Study Selection

The methodology has been described elsewhere [16].

### 2.5. Quality Assessment

Quality Assessment Tool of the National Institutes of Health [17] was used. The methodology has been described elsewhere [18].

### 2.6. Definitions

An implant was considered a failure if presenting signs and symptoms that led to implant removal, i.e., a lost implant. Implant failure could be either early (the inadequacy of the host to establish or promote osseointegration in the early stages of healing) or late (the failure of either the established osseointegration or function of dental implants) [19].

Diabetes mellitus was defined according to the International Diabetes Federation [2] as “a long-term condition that occurs when raised levels of blood glucose occur because the body cannot produce any or enough of the hormone insulin or cannot effectively use the insulin it produces.” Diabetes type 1 applies to the cases when the body produces very little or no insulin, and diabetes type 2 for the cases when there is an inability of the body’s cells to respond fully to insulin (insulin resistance) [2].

Information on the number of implants/patients among the different types of diabetes mellitus (type 1 and 2) was collected as reported by the authors of the publications.

MBL was defined as loss, in an apical direction, of alveolar bone marginally adjacent to the dental implant, in relation to the marginal bone level initially detected after the implant was surgically placed. Only studies using the long-cone parallel technique for periapical radiographs were considered.

### 2.7. Data Extraction

The methodology has been described elsewhere [20].

### 2.8. Analyses

The methodology for meta-analysis has been described elsewhere [16,18,20]. Meta-regressions were performed to verify how the odds ratio (OR) and MBL were associated with the time of follow-up. The data were analyzed using OpenMeta [Analyst] [21]. A funnel plot (plot of effect size versus standard error) was drawn with the software OpenMEE [22].

## 3. Results

### 3.1. Literature Search

The study selection process is summarized in Figure 1. The search initially resulted in 1471 papers (193 in Pubmed, 259 in Web of Science, 1019 in ScienceDirect—in the last one, the filter ‘Article type—Research articles’ was used due to the great number of initial entries), of which 89 publications were eligible for inclusion (see Appendix A for list of included articles).

### 3.2. Description of the Studies

Detailed data on the 89 included studies published between 1999 and 2021 are shown in Appendix A. Studies were either unicenter (*n* = 73) or multicenter (*n* = 16). Eight studies were randomized clinical trials (RCT), 11 prospective studies (without a pre-established controlled group), 15 were prospective controlled clinical trials, and 55 were retrospective observational studies. Countries where the studies were more often conducted (other countries could be included in case of multicenter studies) included the USA (*n* = 19), Italy (*n* = 13), Spain (*n* = 6), Brazil, Germany, and Belgium (5 studies each), Austria, Portugal, and South Korea (4 studies each), and Sweden (*n* = 3), among others.

The mean follow-up ± standard deviation of 72 studies was 38.8 ± 35.0 months (min–max, 3–194.3). There was no precise information on follow-up time for the other 17 studies; for example, “patients were followed up between the years 2006 to 2009”, or “patients were followed up for up to 48 months”.

Different loading protocols were used in the studies, with delayed loading being the most common (43 studies), followed by immediate loading (34 studies), early loading (4 studies), and not prosthetic loaded (4 studies). This information was not available in 23 studies. One of the loading protocols could be applied for all implants of a study or a combination of them for different implants of the same study.

Sixty-three studies included implants installed in both jaws, 13 studies included patients with implants only in maxillae, and the 13 studies included patients with implants only in mandibles.

Eight studies did not include smokers among their patients, and information on the presence/absence of smokers in the cohort group was not available in six studies.

Altogether, there were 5510 implants (394 failures) placed in diabetic patients and 62,780 implants (2343 failures) placed in non-diabetic patients. Implants from the following manufacturers were most often used in the studies: Nobel Biocare (Göteborg, Sweden) in 39 studies, Straumann (Basel, Switzerland) in 20 studies, and Astra Tech (Mölndal, Sweden) in 11 studies. Information on which implant brand and/or system was used was not available in 12 studies.

A comparison of the mean MBL between diabetic and non-diabetics was reported in 10 studies, of which 9 also provided information on the standard deviation, which is necessary to conduct a meta-analysis of continuous variables.

### 3.3. Quality Assessment

All included studies were classified as “good” (Appendix A). In most cases, the main issues in the publications were related to not well-described statistical methods and to the inclusion of non-consecutive patients in the studies.

### 3.4. Meta-Analyses

A random-effects model was used to evaluate the comparison of the implant failure between the two groups, due to heterogeneity (τ^2^ = 0.721, Chi^2^ = 224.856, I^2^ = 60.864, *p* < 0.000).

Implants placed in diabetic patients had a higher risk of failure than implants placed in non-diabetic patients, with an OR of 1.777 (95% CI, 1.344, 2.352, *p* < 0.001; Figure 2), meaning that diabetic patients presented a 1.777 higher risk to lose an implant than non-diabetic patients; i.e., implants placed in diabetic patients have a higher risk of failure by 77.7% in relation to the ones placed in non-diabetic patients.

Subgroup analysis for implant failure when only studies evaluating implants inserted in maxillae were pooled resulted in OR of 1.968 (95% CI, 1.031, 3.759, *p* = 0.040; Figure 3, and an OR of 1.805 (95% CI, 0.911, 3.575, *p* = 0.090; Figure 4) for when only studies evaluating implants inserted in mandibles were pooled. Thus, the difference in implant failure between the groups was statistically significant in the maxilla but not in the mandible.

A sub-analysis for the group of studies providing information on implant failures between patients with diabetes mellitus type 1 and type 2 was also performed, resulting in an OR of 4.477 (95% CI, 1.134, 17.676, *p* = 0.032; Figure 5).

The MD of MBL between the groups was 0.776 mm (95% CI, 0.090, 1.461, standard error 0.350, *p* = 0.027) (τ^2^ = 1.217, Chi^2^ = 6083.123, I^2^ = 99.852, *p* < 0.001) (Figure 6), meaning that implants placed in diabetic patients presented a mean 0.776 mm higher MBL than the implants placed in non- diabetic patients. The difference was statistically significant.

### 3.5. Meta-Regressions

Information on the (mean) follow-up time was available in 72 publications, while no precise information on follow-up (for example, life-table or Kaplan–Meier analysis) was available for the remaining 17 studies.

In a meta-regression including these 72 studies, it was observed that the follow-up time had an effect on the OR of implant failure between the groups (Figure 7), resulting in the following linear equation:

y = 0.922 − 0.007x, where:

Intercept = 0.922 (0.515, 1.329), standard error 0.208, *p* < 0.001

Follow-up = −0.007 (−0.014, 0.000), standard error 0.003, *p* = 0.048

There was an estimated decrease of 0.007 in OR for every additional month of follow-up, with statistical significance.

A sensitivity analysis of the meta-regression plotting together only the studies with follow-up up until 5 years (Figure 8) resulted in the following linear equation:

y = 1.117 − 0.015x, where:

Intercept = 1.117 (0.529, 1.705), standard error 0.300, *p* < 0.001

Follow-up = −0.015 (−0.034, 0.003), standard error 0.010, *p* = 0.105

In this case, there was an estimated decrease of 0.015 in OR for every additional month of follow-up, although not statistically significant.

A meta-regression considering the effect of follow-up on MBL mean difference between groups (Figure 9) resulted in the following first-degree equation:

y = −0.510 + 0.032x, where:

Intercept = −0.510 (−1.320, 0.301), standard error 0.414, *p* = 0.218

Follow-up = 0.032 (0.015, 0.049), standard error 0.009, *p* < 0.001

There was an estimated increase of 0.032 mm in the mean difference of MBL between groups for every additional month of follow-up, with statistical significance.

### 3.6. Publication Bias

The funnel plot did not show a clear asymmetry (Figure 10), indicating a possible absence of publication bias.

## 4. Discussion

The aim of the present systematic review was to compare the clinical outcomes of dental implants between diabetic and non-diabetic patients. This is not the first review on the subject. However, previous reviews either failed to conduct any statistical analysis [23] or were based on much fewer clinical studies [14,24,25]. The present review adds much more data (from 89 studies) for the analyses and is the first one in many aspects: (a) to perform a sub-analysis comparing dental implant failure rates between type 1 and type 2 diabetic patients; (b) to perform subgroup analyses for implant failure when only studies evaluating implants inserted in maxillae, as well as when only studies evaluating implants inserted in mandibles; (c) to perform a meta-regression testing the association between the odds ratio of implant failure between diabetic and non-diabetic individuals, and the follow-up time; (d) to perform a meta-analysis on the difference of MBL between diabetic and non-diabetic patients; and (e) to perform a meta-regression testing the association between follow-up and the MBL mean difference between diabetic and non-diabetic individuals.

According to the results of the present review, diabetic patients presented a statistically significant higher risk of dental implant failure and higher marginal bone loss than non-diabetic patients. The null hypothesis was therefore rejected. These results are thought to be mainly related to the deleterious effects of diabetes mellitus on many physiological processes in the human body.

One of the negative effects of diabetes mellitus on the body is impaired bone metabolism and bone strength. The hyperglycemia associated with diabetes mellitus, usually due to poor glycemic control, may worsen bone mineral density (BMD), along with an increased risk of fractures. This is caused by an increase in urinary calcium excretion and by the accumulation of advanced glycation products, which induces a proinflammatory state, resulting in lower insulin-like growth factor 1 (IGF-1) levels, and lower pH/acidosis [26]. The role of IGF-1 is important, as it increases bone matrix synthesis and bone formation, as well as regulates osteoclastogenesis by promoting their differentiation [27]. A clinical study observed that patients with diabetes mellitus type 1 had a lower total body bone mineral density as compared to age, sex, and body mass index and matched non-diabetic controls [28].

Another damaging effect of the disease is the delayed wound and bone healing. The placement of a dental implant into the jaws is controlled surgical aggression to the bone tissues. The healing around the installed implant begins with the formation of a blood clot, vascularization, and proliferation and migration of mesenchymal stem cells (MSCs) from surrounding bone marrow [29]. Under favorable conditions and stable sites, MSCs differentiate into osteoblasts, and woven bone forms through osteogenesis followed by compaction of woven bone, and after a period of time, bone remodeling starts [30]. Anything that could impair this process may jeopardize the osseointegration of a dental implant. In diabetic patients, the impaired bone cell metabolism and subsequent changes in the properties of the bone matrix may contribute to undermining proper bone healing and reducing the bone matrix strength [31].

It is known that diabetes mellitus causes microvascular complications. When exposed to hyperglycemia, some types of capillary endothelial cells are unable to reduce the transport of glucose inside the cell, which makes these cells more likely to become damaged as a result of constant hyperglycemia inside them [32]. Several hypotheses have then been proposed to explain the biochemical process of developing microvascular complications (for details, check [33]). The issue may very probably affect the survival of dental implants, as their clinical success is dependent not only upon osseointegration but also on neovascularization in the peri-implant bone [34], and since neoangiogenesis is not possible without the development of new blood vessels from pre-existing vasculature, involving the migration behavior, proliferation and differentiation of endothelial cells [35], damage of pre-existing capillary endothelial cells may very well have a negative effect on the clinical outcomes of dental implants.

Hyperglycemia in diabetes mellitus causes dysfunction of the immunological response through many mechanisms, which include suppression of cytokine production (cytokines induce the innate immune response, inflammation, and the adaptive immune response) [36], phagocytosis impairment [37], inhibition of complement effectors [38], dysfunction of immune cells [39], and reduced leukocytes recruitment [40]. Therefore, diabetic individuals are more susceptible to infections [7]. This may have a considerable influence on the long-term survival of dental implants, as the immune system is needed to tackle the stages of bacterial establishment and infection of the peri-implant tissues [41].

All these factors may directly or indirectly impair the osseointegration process and/or the long-term maintenance of dental implants in the jaws.

The dysfunction of the immunological response, together with the delayed wound healing, may have some influence on the significantly higher MBL around implants in diabetic than in non-diabetic patients, as observed in the present results. The results of an animal-model study suggested that hyperglycemia can be associated with bone loss around implants [42], possibly related to the increased levels and accumulation of advanced glycation end products in the gingival tissue [43], which in turn triggers osteoclast induction and promote bone resorption [44]. The results of a review on the subject suggested that elevated and poorer glycemic levels are associated with a greater prevalence of peri-implantitis [45]. Moreover, higher HbA1c levels have been associated with greater MBL [46]. The estimated increase in the mean difference of MBL between diabetic and non-diabetic patients may be a reflection of the cumulative deleterious effects of the disease with time [47]. This suggests that a closer control of peri-implant tissues may be necessary for diabetic patients in comparison to non-diabetic patients. A review on the effect of the treatment of periodontal disease for glycemic control in diabetic patients concluded that there was no evidence to support that one periodontal therapy was more effective than another in improving glycemic control in people with diabetes mellitus [48], although the review focused on periodontitis, not peri-implantitis. A review of the impact of diabetes on oral bone regeneration and augmentation techniques concluded that the level of evidence about it is still low [49].

The possible impact of different implant surfaces on MBL in diabetic and non-diabetic patients is something important to be considered. This would be easier if there were more data available in order to conduct comparisons between different surface modifications. However, a limited number of studies provided information on mean MBL with standard deviation. Therefore, an attempt to conduct additional sub-group analyses of MBL by different implant surfaces would not result in any reliability and would mislead the interpretation of the data. Although more recent surface treatments have shown improvements in the bone-implant contact, it is not entirely clear whether, in general, one surface modification is better than another [50], and although surface modifications of modern dental implants may result in less MBL than those surfaces from implants from the 1990′s [51], this is not always the case [52].

According to the present results, there was a statistically significant difference in the failure rate between the diabetic and non-diabetic patients for implants placed in the maxilla but not in the mandible. This could be related to the fact that sites with poorer bone quality and lack of bone volume, which are more common in the upper jaw, may negatively affect the implant failure rates [53].

There was an estimated decrease in OR for every additional month of follow-up, meaning that the difference in the risk of implant failure between diabetic and non-diabetic patients tended to decrease with time slowly. Even with the statistically significant difference in failures between the groups, this could be related to the higher implant failure rate usually observed within the first year after implant installation, regardless of how long the follow-up might be [54,55].

The sub-analysis comparing the failure rates between patients with different types of diabetes mellitus suggested that patients with diabetes mellitus type I are much more likely to lose an implant than patients with type 2 of the disease. Although these results are based on limited data, some complications associated with the disease are worse in type 1 diabetes than in type II, which may support these results. Type 1 and type 2 diabetes mellitus are heterogeneous diseases, and their progression and clinical presentation may vary to a great extent. Most cases of type 1 diabetes mellitus are caused by cellular-mediated autoimmune destruction of the pancreatic β-cells, with a minority of cases with no known etiologies [13]. The pancreatic β-cells synthesize, store, and release insulin, in order to maintain the circulating glucose concentrations within a physiologically acceptable range [56]. As these cells are destroyed, excessive levels of glucose must be dealt with exogenous insulin in type 1 diabetic patients. Patients with type 2 diabetes have a relative insulin deficiency, and there is peripheral insulin resistance with progressive loss of β-cell adequate function [57,58]. These differences in the pathophysiology between the disease types, together with poorer adherence to treatment regimens [59] and greater difficulty in achieving metabolic control [60] in type 1 diabetes, have an influence on the severity of symptoms, which is often marked in type 1 diabetic patients. Although the severity may vary in type 2 diabetic patients, it is usually not severe in these individuals [13]. Diabetes mellitus type 1 usually has an earlier onset, resulting in earlier development of micro- and macro-vascular complications in comparison to type 2 diabetes mellitus [61]. Moreover, patients with type 1 diabetes usually present early bone loss, whereas in type 2, the development of abnormal osseous architecture results in increased or normal bone mineral density [62], although with compromised skeletal quality and strength [63]. All this may result in a more compromised implant and bone site in type 1 diabetic patients than in type 2 patients. Individuals with type 1 diabetes mellitus even have a higher loss of life expectancy than those with type 2 [61] due to the relatively higher incidence of cardiovascular diseases and acute metabolic disorders in type 1 diabetes mellitus [64].

### Limitations of the Present Study

The limitations of the present results include the fact that (a) many included clinical studies were retrospective trials; (b) many studies have a small sample size as well as a short follow-up, which in turn can lead to an underestimation of the failure rates; (c) several studies did not aim to compare clinical outcomes between diabetic and non-diabetic patients; (d) the clinical outcomes could have been affected by many confounding factors. Moreover, individuals may present multiple risk factors [65,66]. It is difficult to estimate the impact of these factors on the outcomes if these variables are not identified separately between diabetic and non-diabetic patients.

## 5. Conclusions

In conclusion, implants placed in diabetic patients present a statistically significant higher risk of failure and greater marginal bone loss than implants placed in non-diabetic patients. When it comes to the comparison between different types of diabetes mellitus, implants placed in diabetic type I patients present a much higher risk of failure than implants placed in diabetic type II patients.

## Figures and Tables

**Figure 1 materials-15-03227-f001:**
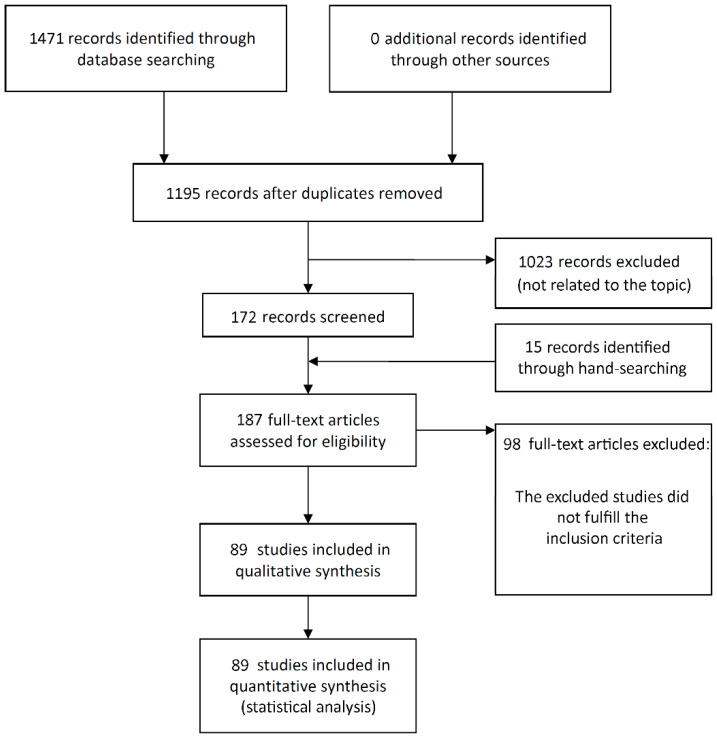
Study screening process.

**Figure 2 materials-15-03227-f002:**
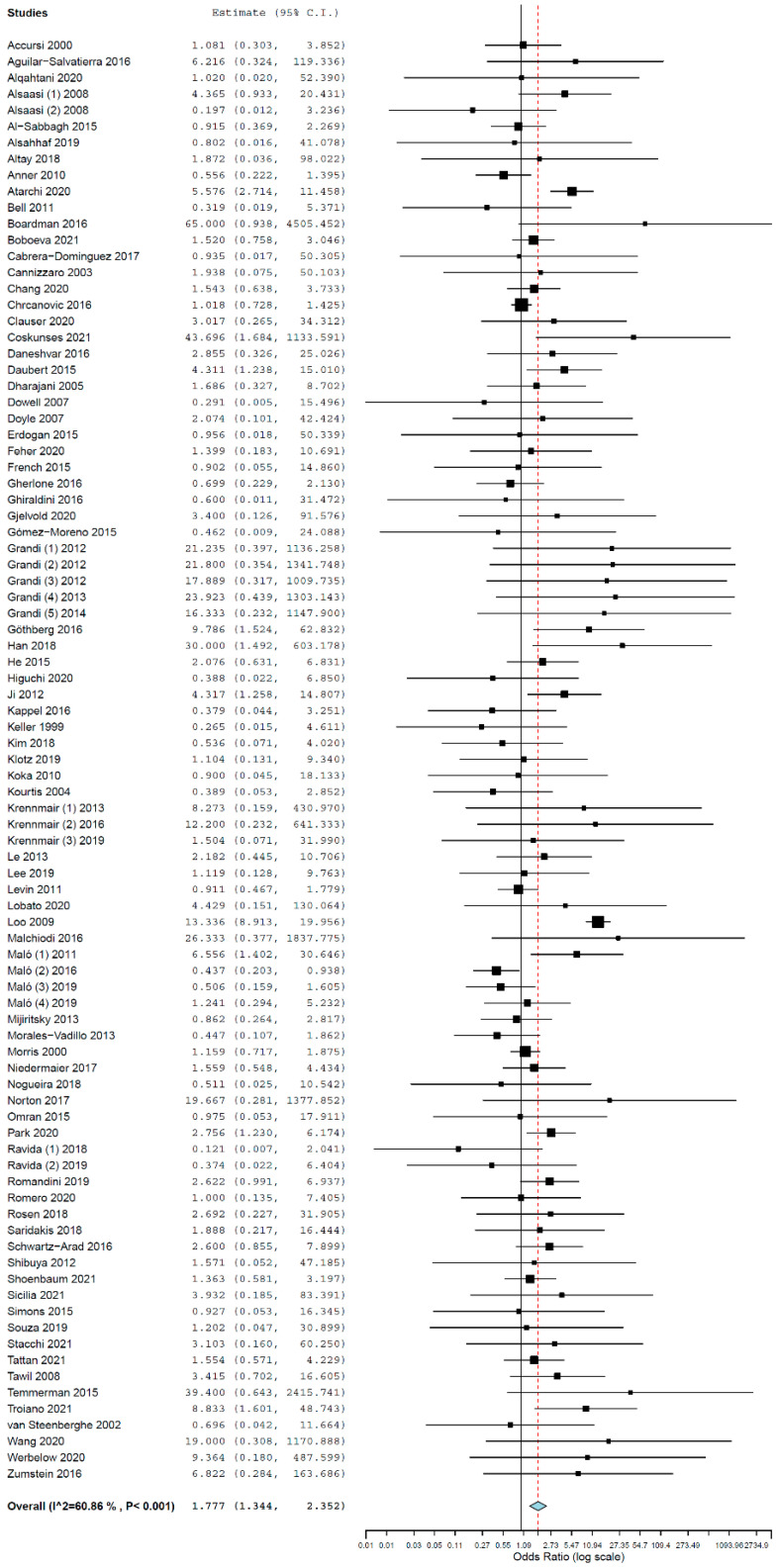
Forest plot for the event ‘implant failure’, global results.

**Figure 3 materials-15-03227-f003:**
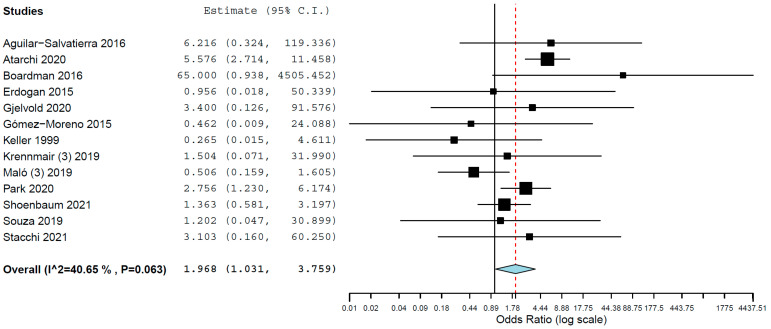
Forest plot for the event ‘implant failure’, studies evaluating implants inserted exclusively in maxillae.

**Figure 4 materials-15-03227-f004:**
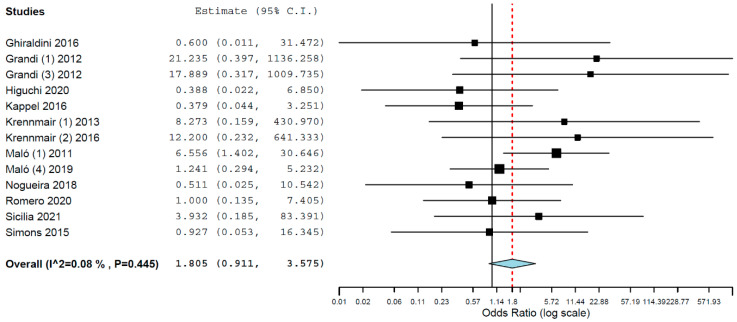
Forest plot for the event ‘implant failure’, studies evaluating implants inserted exclusively in mandibles.

**Figure 5 materials-15-03227-f005:**
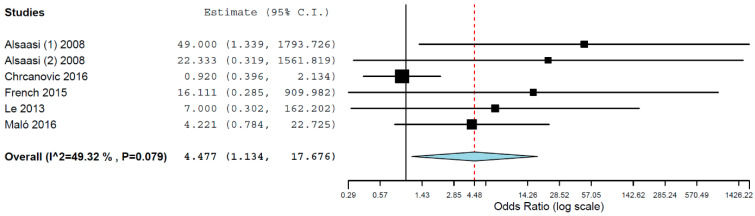
Forest plot for the event ‘implant failure’ between diabetes mellitus type I and type II.

**Figure 6 materials-15-03227-f006:**
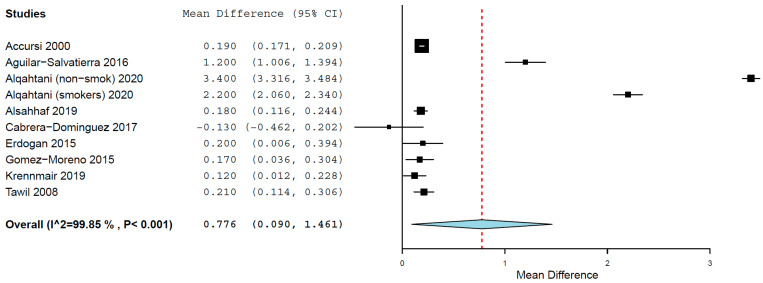
Forest plot for the event ‘marginal bone loss’.

**Figure 7 materials-15-03227-f007:**
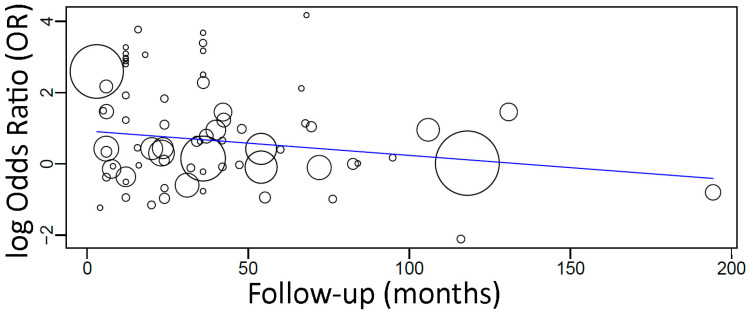
Scatter plot for the meta-regression with the association between the odds ratio (OR) of implant failure between diabetic and non-diabetic individuals, and the follow-up time (in months). Every circle represents a study and the size of the circle represents the weight of the study in the analysis.

**Figure 8 materials-15-03227-f008:**
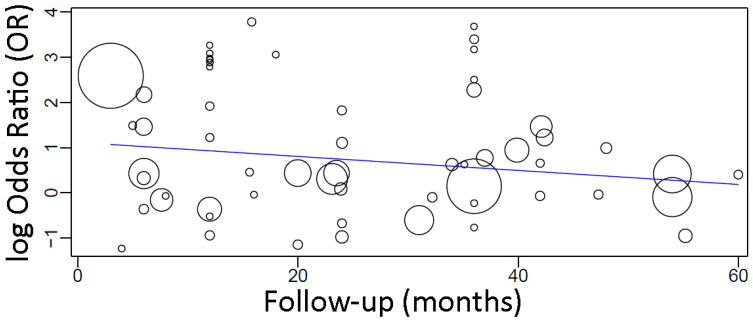
Scatter plot for the meta-regression with the association between the odds ratio (OR) of implant failure between diabetic and non-diabetic individuals, and the follow-up time (in months; limited to 60 months). Every circle represents a study and the size of the circle represents the weight of the study in the analysis.

**Figure 9 materials-15-03227-f009:**
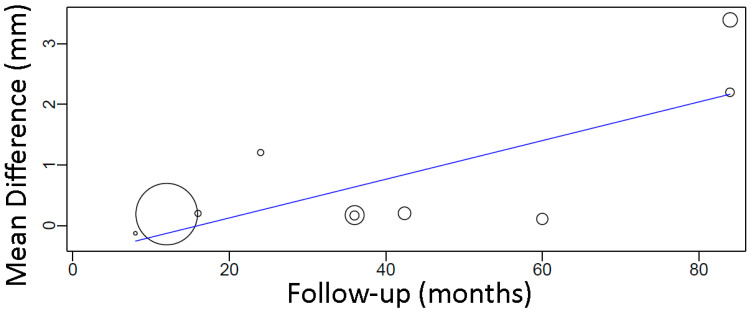
Scatter plot for the meta-regression with the association between follow-up (in months) and MBL mean difference between diabetic and non-diabetic individuals. Every circle represents a study and the size of the circle represents the weight of the study in the analysis.

**Figure 10 materials-15-03227-f010:**
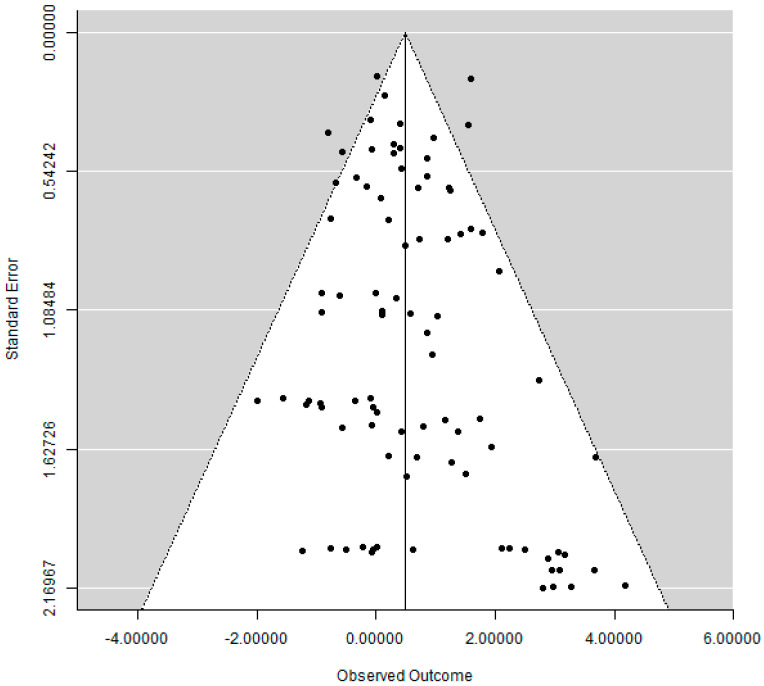
Funnel plot.

## Data Availability

The data presented in this study are available within the article and Appendix A.

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
