# Peer review of "Diabetes Mellitus and Dental Implants: A Systematic Review and Meta-Analysis"

_materials, 2022, doi:10.3390/ma15093227_

Round 1
Reviewer 1 Report
In the manuscript "Diabetes mellitus and dental implants: a systematic review and 2 meta-analysis" authors point to implants placed in diabetic patients present statistically significant higher risk of failure and greater marginal bone loss than implants placed in non-diabetic patients. The paper is written clearly and clearly, and all the results obtained were adequately commented on in the discussion. The authors clearly state the shortcomings of the analyzed studies. one of the disadvantages is the presence of variables other than diabetic and non-diabetic on the implant placement outcomes and one of them is smoking. Do the authors of the processed clinical studies list other risk factors and if so, it would be good to show them in the table.
Author Response
Question 1: In the manuscript "Diabetes mellitus and dental implants: a systematic review and meta-analysis" authors point to implants placed in diabetic patients present statistically significant higher risk of failure and greater marginal bone loss than implants placed in non-diabetic patients. The paper is written clearly and clearly, and all the results obtained were adequately commented on in the discussion. The authors clearly state the shortcomings of the analyzed studies.
Reply: Thank you very much for your time dedicated to review our manuscript.
Question 2: One of the disadvantages is the presence of variables other than diabetic and non-diabetic on the implant placement outcomes and one of them is smoking. Do the authors of the processed clinical studies list other risk factors and if so, it would be good to show them in the table.
Reply: The reviewer has made an important observation. We tried to collect as many confounding factors as possible, such as study design, number of male/female patients, patients’ age range, prosthetic loading protocol, the presence of different types of diabetes, verification of blood sugar level, implant location, implant system/model used, and the presence of smokers. However, most of the included studies were retrospective in nature, resulting in gaps in information, and the great majority of the them did not provide too much information on other confounding factors than the aforementioned ones. You can notice that even for smoking, a well-known factor to affect implant outcomes, was the information not available (or clearly available) in all studies. Therefore, information on any other additional risk factor than the ones already mentioned will be very fragmented.
Reviewer 2 Report
Authors presented systematic review on diabetes mellitus and dental implants. Following points are important and needed to be incorporated.
- Introduce problem statement in abstract.
- Introduction section is not comprehensive. Authors should try to include reasonable detail on the topics involved in the field.
- Systematic review is nice to get an idea about what is happening in the field overall. But only presenting key information without debating on the fundamental science, application areas, comparison of different studies, introduction to field and sub topics, it is not adding desired value to readership.
Authors should include comprehensive information on the topic along with discussion on analysis. Currently, the paper looks like a report. Good luck to authors.
Author Response
Question 1: Authors presented systematic review on diabetes mellitus and dental implants.
Reply: Thank you very much for your time dedicated to review our manuscript.
Question 2: Introduce problem statement in abstract.
Reply: We agree with the reviewer, However, according to the journal’s instructions for authors, “The abstract should be a total of about 200 words maximum.” And the current abstract already contains 197 words. In order to introduce the problem statement in the abstract, we would have to remove some other information from it. We don’t know how we would delete any of the information already written in the abstract without removing the reporting of important results that would compromise the short message of an abstract to the readers.
Question 3: Introduction section is not comprehensive. Authors should try to include reasonable detail on the topics involved in the field.
Reply: We ask the reviewer to please specify which “topics involved in the field” would be, if the focus of the review was to investigate the possible impact of diabetes on the dental implant outcomes. Please let us know.
Question 4: Systematic review is nice to get an idea about what is happening in the field overall. But only presenting key information without debating on the fundamental science, application areas, comparison of different studies, introduction to field and sub topics, it is not adding desired value to readership.
Reply: The present review focused on the possible impact of diabetes on the dental implant outcomes. The application area would be, therefore, Odontology/Dentistry. Introduction to the field was provided with updated epidemiological information on diabetes (in the Introduction), as well as the definition of the condition (in the Materials and Methods), and what characterizes diabetes and its possible consequences (in the Introduction). A comparison of different studies concerning many variables was already presented in Table S1, that the reader can find in the Supplementary Material. And the “fundamental science”, which I would call the biochemical mechanisms behind the disease, was already presented in the paragraphs 2, 3, 4, 5 and 7 of the Discussion section.
As already mentioned, the present review did not focus on too much detailed information about diabetes, but on the possible impact of diabetes on the dental implant outcomes. And we introduced the reader and provided information on many sub-topics on diabetes. It is possible for any reader to go deeper (deeper than the information already provided) into the subject “diabetes mellitus” and/or its sub-topics by consulting the text of the quoted references in the manuscript, or specialized books on the subject. The debate here is focused on the research question, which is well described under “Objective” in the Materials and Methods section. Anything else would go beyond the focus of the review.
Question 5: Authors should include comprehensive information on the topic along with discussion on analysis. Currently, the paper looks like a report. Good luck to authors.
Reply: Comprehensive information on diabetes the reader can find by consulting the text of the quoted references in the manuscript, or specialized books on the subject. The focus of the review was to investigate the possible impact of diabetes on the dental implant outcomes. In the Discussion section we tried to explain the possible reasons behind the findings of the study based on the already known knowledge about diabetes. Anything beyond that would make the article look like a book chapter on diabetes, which is not the point of this review. Moreover, as already mention in our response to your previous observation, introduction to the field was provided in the Introduction with updated epidemiological information on diabetes, as well as the definition of the condition (in the Materials and Methods), what characterizes diabetes and its possible consequences (in the Introduction), and the biochemical mechanisms behind the disease, was already presented in the paragraphs 2, 3, 4, 5 and 7 of the Discussion section. A much deeper information about all these topics, which was not the aim of the present review, can be found in more specialized literature. Otherwise, we ask the reviewer to please be more specific about which kind of information he/she would like to see in the text.
Reviewer 3 Report
This review sought to assess the impact of diabetes mellitus on dental implant failure rates and marginal bone loss. The authors wrote an interesting detailed review with evidence, but there are several issues that I am concerned about.
Introduction
There are several similar systematic review such as "Dubey RK, Gupta DK, Singh AK. Dental implant survival in diabetic patients; review and recommendations. Natl J Maxillofac Surg. 2013 Jul;4(2):142-50. doi: 10.4103/0975-5950.127642. PMID: 24665167; PMCID: PMC3961886." and "Naujokat, H., Kunzendorf, B. & Wiltfang, J. Dental implants and diabetes mellitus—a systematic review. Int J Implant Dent 2, 5 (2016). https://doi.org/10.1186/s40729-016-0038-2"
It is important to mention the advantages of the current study compared with the literature.
Materials and Methods
The methods is well-organized and clear. But there are some minor issues: 1. in the inclusion criteria, the type of included studies is not mentioned (for instance, did you include case series?)
2. Reviewing process is not clear (number of reviewers, etc.)
Discussion
There are two major points to discuss more here. 1. the impact of immunity on periodontal diseases is not completely explained here. 2. The other risk factors that might not match or be considered in the reviewed articles such as smoking, gender, etc.
References
There are some minor errors (Ref#10, 49, etc.). Recheck the references.
Author Response
Question 1: This review sought to assess the impact of diabetes mellitus on dental implant failure rates and marginal bone loss. The authors wrote an interesting detailed review with evidence, but there are several issues that I am concerned about.
Reply: Thank you very much for your time dedicated to review our manuscript.
Question 2: There are several similar systematic review such as "Dubey RK, Gupta DK, Singh AK. Dental implant survival in diabetic patients; review and recommendations. Natl J Maxillofac Surg. 2013 Jul;4(2):142-50. doi: 10.4103/0975-5950.127642. PMID: 24665167; PMCID: PMC3961886." and "Naujokat, H., Kunzendorf, B. & Wiltfang, J. Dental implants and diabetes mellitus—a systematic review. Int J Implant Dent 2, 5 (2016). https://doi.org/10.1186/s40729-016-0038-2"
It is important to mention the advantages of the current study compared with the literature.
Reply: Indeed. The advantages of the current review in relation to previous ones were discussed in the first paragraph of the Discussion section.
Question 3: The methods is well-organized and clear. But there are some minor issues: 1. in the inclusion criteria, the type of included studies is not mentioned (for instance, did you include case series?)
Reply: By “clinical human studies” is already implied that case series and cohort studies were considered. Moreover, it was later mentioned in the text that non-randomized and retrospective clinical studies were also considered, not only RCT, as it is not possible to randomize the placement of implants for diabetes, namely, a person is either diabetic or not.
Question 4: Reviewing process is not clear (number of reviewers, etc.)
Reply: Except for the fact the study selection process was conducted by the three authors of the review (and now this information was included in the text), which other information the reviewer would like to have explained in a clearer way? Could you please be more specific? We wrote the objectives of the study, the search strategies, the inclusion and the exclusion criteria, the study selection process, the quality assessment tool used, the definitions, the data extraction process, and the parameters for the meta-analyses and the meta-regressions. So, which factors need to be clearer? Could be reviewer please point them out?
Question 5: There are two major points to discuss more here. 1. the impact of immunity on periodontal diseases is not completely explained here.
Reply: This is also a factor to possibly influence periodontal status, and it was already discussed in the 5th paragraph of the Discussion section:
“Hyperglycemia in diabetes mellitus causes dysfunction of the immunological response through many mechanisms, which include suppression of cytokine production (cytokines induce the innate immune response, inflammation and the adaptive immune response) [28], phagocytosis impairment [29], inhibition of complement effectors [30], dysfunction of immune cells [31], and reduced leukocytes recruitment [32]. Therefore, diabetic individuals are more susceptible to infections [5]. This may have a considerable influence on the long-term survival of dental implants, as the immune system is needed to tackle the stages of bacterial establishment and infection of the peri-implant tissues [33].”
However, the present review focused on the possible impact of diabetes on the dental implant outcomes. To go too deep into the subject would go beyond the aim of the present review. A much deeper information about it can be found in more specialized literature, and the readers can do so by consulting the cited references in this paragraph, as well as to search for it in health-related electronic databases, such as Pubmed/Medline.
Question 6: 2. The other risk factors that might not match or be considered in the reviewed articles such as smoking, gender, etc.
Reply: We agree that confounding factors are of great importance. However, the real fact is that individual patients sometimes present with more than one risk factor, and groups of patients are typically heterogeneous with respect to risk factors and susceptibilities so the specific effect of an individual risk factor could be isolated neither for individual studies nor for the present review. This is understandable and expected because study populations are typically representative of normal populations with various risk factors. To precisely assess the effect of a risk factor on implant outcomes, it would be ideal to eliminate all other risk factors from the study population. Not only does the coexistence of multiple risk factors within a study population create an inability to assess the specific effect of one individual risk factor, but there is a possibility that certain risk factors together may be more detrimental than the individual risk factors alone. Therefore, not having separated information on several confounding factors between diabetic and non-diabetic patients would it make impossible to draw any conclusion on the possible effect of these other factors, taking into consideration the dichotomous analysis between diabetics vs. non-diabetic patients in the present review. The possible influence of several factors on dental implant outcomes is well-investigated in the literature. However, to only deeply discuss them (besides what was already written under “Limitations of the present study” in the Discussion section) without being able to analyze them in the context “diabetic vs non-diabetic” would go beyond the aim of the present review.
Question 7: There are some minor errors (Ref#10, 49, etc.). Recheck the references.
Reply: All references are OK. We used EndNote to manage the references, with the template provided by MDPI, which is already set to fit the reference guidelines of the journal. We cannot modify the template provided by the publisher. If you meant that “Bmj” should be “BMJ” instead, this was corrected.
Round 2
Reviewer 2 Report
The review still lacks focus on following areas comprehensively. Introduce state of the arts and fundamental science of following sample topics (not limited to), you can introduce more as per the focus of study.
- Diabetes and osseointegration
- Diabetes and peri-implantitis
- Diabetes and implant survival
- Diabetes and bone augmentation
- Influence of duration of diabetes disease
- Influence of supportive therapy
- Influence of quality of glycemic control
These sample topics are from literature, add comprehensive introduction and discussion on above or more topics. This will enrich the quality and give readers comprehensive background along with current status of the field.
Good luck
Author Response
We have added new text concerning the listed sub-topics (even though some of them were already well discussed in the Discussion), although this was not done in a ‘comprehensively’ way, as comprehensive means “including or dealing with all or nearly all elements or aspects of something”, and our review focused on the possible influence of diabetes on some clinical outcomes of dental implants, namely implant failure and marginal bone loss. If we would cover all these sub-topics comprehensively, the manuscript would become a book or a book chapter, which was not the aim of the present review. Thus, in order to have a comprehensive understanding about these sub-topics the reader will have to turn to other literature, with includes some references cited in the manuscript, as well as books on the subject.
Regarding the sub-topics:
1) Diabetes and osseointegration: The possible reasons why diabetes may influence the osseointegration process and the biochemical/immunological processes behind it had already been well discussed in paragraphs 3 to 6 of the Discussion, ending up with paragraph 7: “All these factors may directly or indirectly impair the osseointegration process and/or the long-term maintenance of dental implants in the jaws.”
2) Diabetes and peri-implantitis: This subject had already been mentioned in the eighth paragraph of the Discussion. We added some addition information to it.
3) Diabetes and implant survival: The investigation of this was one of the main aims of the present review. The present review itself represents the most updated information about it, and the possible reasons why diabetes may influence the survival of dental implants was already well discussed in paragraphs 3 to 7 of the Discussion section, as well as in paragraph 12 of the Discussion, focused on the suggested difference of failure rates between type 1 and type 2 diabetic patients.
4) Diabetes and bone augmentation: Some information about it was added to the eighth paragraph of the Discussion.
5) Influence of duration of diabetes disease: This subject had already been mentioned in the second paragraph of the Introduction. We added some addition information to it.
6) Influence of supportive therapy: Some information about it was added to the eighth paragraph of the Discussion.
7) Influence of quality of glycemic control: Although the third paragraph of the Introduction had already introduced the reader to this topic, we added some additional information to it.